# Dilation-Erosion Methods for Radiograph Annotation in Total Knee Replacement

**Yehyun Suh**[1,2]                                                        YEHYUN.SUH@VANDERBILT.EDU

**Aleksander Mika**[3]                                                 ALEKSANDER.MIKA@VUMC.ORG

**J. Ryan Martin**[3]                                                       JOHN.MARTIN@VUMC.ORG

**Daniel Moyer**[*1,2]                                               DANIEL.MOYER@VANDERBILT.EDU

[1] *Department of Computer Science, Vanderbilt University, Nashville, TN, USA*

[2] *Vanderbilt Institue for Surgery and Engineering, Nashville, TN, USA*

[3] *Department of Orthopaedic Surgery, Vanderbilt University Medical Center, Nashville, TN, USA*

## Abstract

In the present work we describe a novel training scheme for automated radiograph anno-tation, as used in post-surgical assessment of Total Knee Replacement. As we show exper-imentally, standard off-the-shelf methods fail to provide high accuracy image annotations for Total Knee Replacement annotation. We instead adopt a U-Net based segmentation style annotator, relax the task by dilating annotations into larger label regions, then pro-gressively erode these label regions back to the base task on a schedule based on training epoch. We demonstrate the advantages of this scheme on a dataset of radiographs with gold-standard expert annotations, comparing against four baseline cases.

**Keywords:** Label Augmentation, X-Ray, Landmark Annotation

## 1. Introduction

Total Knee Replacement (TKR) is a standard therapy for advanced knee arthritis (Kim et al., 2021; Evans et al., 2019). Post-surgical evaluation of TKR relies partially on radio-graphs of the patient's knee and implant, and the alignment of that implant to the femur and tibia. These assessments are made by the manual placement of markers by orthopedic clinicians, and may be made at regular outpatient follow-up visits in the months after the procedure. In particular, measurements of implant alignment during recovery and later regular use are decision criteria for possible additional correction procedures (in the event of adverse implant positioning/alignment). Automation of this marker placement is a clear target for learned medical vision systems. The benefits of automated marker placement for TKR are also clear: assessment of radiographs without expert intervention, possibly for in-the-field point-of-care assessment, or for reducing assessment loads when assessing retrospective studies of large databases.

However, application of off-the-shelf models results in sub-standard performance. We show that neither direct regression using convolutional architectures(Krizhevsky et al., 2012; He et al., 2016), nor pre-trained convolutional networks (with fine-tuning), nor conventional U-Net(Ronneberger et al., 2015) methods produce acceptable accuracy when applied to TKR annotation tasks. We instead propose a Dilation/Erosion label augmentation method and corresponding training schedule which improves performance of a U-Net based method, taking high baseline errors (100+ pixels) to single-digit pixel errors. We include a short discussion with hypotheses for why standard methods fail, and current directions for further expansion to other Total Knee Replacement radiograph domains.

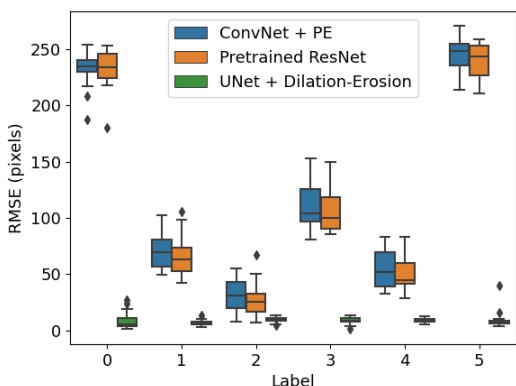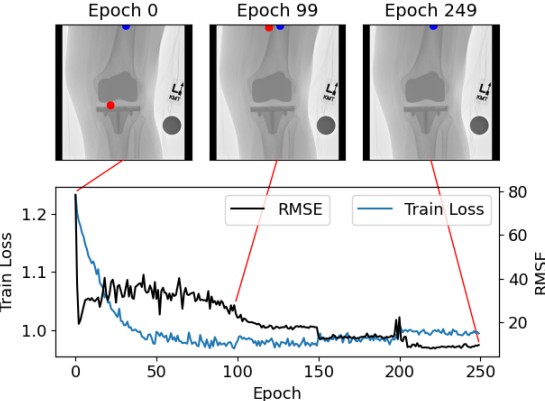

Figure 1: At **Left** we show the RMSE between predicted pixel and the ground truth pixel for labels 0 thru 5 for our test cohort. At **right** we show the training loss (left y-axis) and RMSE (right y-axis) across epochs with exemplar predicted outputs (red) and ground truth labels (blue) for Epoch 0, 99, and 249. Drastic changes in the training loss (e.g., at epoch 150 and 200) are due to the training schedule for erosion and re-weighting.

## 2. Method

Initial attempts at TKR annotation show poor performance (e.g., baselines shown in Section 3). We here construct a dilation-erosion label augmentation method that improves the U-Net methods' performance, using the same general architecture and gradient-based learning. Image labels are first dilated by a set number of image dilation iterations. These dilated labels are allowed to overlap. The prediction network (a U-Net) is trained using the dilated labels. Labels are then eroded over a schedule based on training steps taken.

As we are adjusting the size of each label as training progresses, this leads to changing label imbalance. A common static imbalance solution is to re-weight the error function, biasing predictions away from degenerate solutions. For dynamic imbalances (induced by our training scheduled erosion), we construct a dynamic reweighting scheme, where $w_0$ is a label weight that would have been used had we not performed dilation erosion:

$$\tilde{w} = w \times \frac{\text{input image size} - (\text{number of dilated pixels} + \text{number of label pixels})}{(\text{number of dilated pixels} + \text{number of label pixels})} \quad (1)$$

## 3. Experiments & Results

Our dataset consists of 180 post-operative knee radiographs, which we split into a 162/18 sized training/validation sets. Each image was annotated by a clinician for six anatomic landmarks. Images and the corresponding label masks were padded to a standard size ($512 \times 512$), and histogram normalized to [0,1].

We compared the performance of several baseline methods: a Convolutional Neural Network optionally with Positional Encoding concatenated as input, ResNet101 pretrained

| Experiment | Label0 | Label1 | Label2 | Label3 | Label4 | Label5 | Mean |
|---|---|---|---|---|---|---|---|
| CNN | 227 | 73 | 40 | 115 | 59 | 236 | 149 |
| CNN w/PE | 233 | 72 | 31 | 113 | 55 | 245 | 151 |
| ResNet-101 | 233 | 66 | 27 | 109 | 50 | 240 | 148 |
| Baseline U-Net | 348 | 267 | 234 | 239 | 264 | 376 | 295 |
| D40 | 185 | 18 | 18 | 17 | 15 | 22 | 77 |
| D40 AW | 21 | 15 | 18 | 20 | 17 | 21 | 19 |
| D40 AW ProE 5 | 16 | 8 | **9** | 10 | **2** | 9 | 12 |
| D60 | 20 | 17 | 18 | 29 | 20 | 23 | 23 |
| D60 AW | 26 | 21 | 22 | 18 | 20 | 26 | 24 |
| D60 AW ProE 5 | 15 | 17 | 16 | 17 | 15 | 19 | 17 |
| D60 AW ProE 10 | **9** | **7** | 10 | **9** | 9 | **9** | **10** |

Table 1: We show the mean RMSE per label and mean overall, across validation images, for each method. D## is the number of dilation iterations applied at initialization, AW indicates Adaptive Weighting, and ProE ## is the number of Erosion steps applied every 50 epochs. **Boldface** values indicate best performance by category.

with ImageNet(Deng et al., 2009), and a baseline U-Net architecture. The U-Net architecture was also trained using our proposed Dilation/Erosion method and reweighting scheme. We also train ablations of these proposed methods in order to test the efficacy of each sub-component, and variations of the component parameters.

For training the U-Net architectures we use pixel-wise cross-entropy loss. As a validation metric we use the RMSE of the pixel-wise distance from the most likely pixel (the maximal output logit value) to the groundtruth label position. For training the other baseline methods we use the validation metric (RMSE of pixelwise distance) directly.

As shown in Table 1, in comparison to all baselines both architecturally similar and dissimilar the proposed training method performs well: the mean RMSE across labels decreased from 149 to 10. We plot the per-subject mean error in Figure 1 Left. In Figure 1 Right we show the training dynamics of the proposed method.

## 4. Conclusion & Discussion

In this paper we introduced a method to improve automated radiograph annotations using the Dilation/Erosion method, as well as a custom weighting scheme. We have also shown that direct regression or conventional U-Net methods surprisingly does not perform well in TKR assessment. We hypothesize that this may be due to the relatively small dataset, and the fact that many common augmentations (rotations, some flips, etc.) are impossible since these transformations cannot be applied in a similar way to the label volumes without breaking the knee radiograph context (all knees are imaged from known/chosen angles and views). In future work we plan on expanding our dataset and evaluations to lateral views and annotations, and to assess inter-rater reliability in order to determine the noise ceiling of prediction accuracy.

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
