# OpenReview forum: "Dilation-Erosion Methods for Radiograph Annotation in Total Knee Replacement"
_MIDL.io/2023/Short_Paper_Track — MIDL 2023 Short paper track Poster_

### Official Review · Reviewer_mb3x · 2023-04-15
**interesting method and application**

**Rating:** 8
**Confidence:** 4

**Review:**

Nice paper, cool application

Interesting and novel technical contribution, based on dilating and eroding segmentation labels during the training process

Is of interest to the community

---

### Official Review · Reviewer_r1w5 · 2023-04-21

**Rating:** 5
**Confidence:** 5

**Review:**

This paper proposed a landmark localization approach for post-surgical evaluation of total knee replacement. The approach incorporates dilation and erosion operations and a reweighting scheme in the training process to improve a Unet-based baseline prediction.  Drawbacks of the proposed method is that (1) description of the dilation/Erosion label augmentation method is unclear. The author need to provide more details regarding the dilation and erosion operations; (2) the baseline methods in the experiment are not the state-of-the-art methods, the author may find this paper is useful:

Yao, Qingsong, et al. "One-shot medical landmark detection." Medical Image Computing and Computer Assisted Intervention–MICCAI 2021: 24th International Conference, Strasbourg, France, September 27–October 1, 2021, Proceedings, Part II 24. Springer International Publishing, 2021.

I’d like to suggest the author to compare their method with other state-of-the-art medical landmark localization approaches. In addition, it is better to illustrate what the 6 anatomic landmarks look like.